



# A Comparison of Long-term Trends in Observations and Emission Inventories of NO$_x$

Elena Macdonald[1,2], Noelia Otero Felipe[1], and Tim Butler[1,3]

[1]Institute for Advanced Sustainability Studies, Berliner Straße 130, 14467 Potsdam, Germany
[2]Institute for Environmental Science and Geography, University of Potsdam, Karl-Liebknecht-Str. 24-25, 14476 Potsdam, Germany
[3]Institute for Meteorology, Free University of Berlin, Carl-Heinrich-Becker-Weg 6-10, 12165 Berlin, Germany

**Correspondence:** Tim Butler (tim.butler@iass-potsdam.de)

**Abstract.** Air pollution is a pressing issue that is associated with adverse effects on human health, ecosystems and climate. Despite many years of effort to improve air quality, nitrogen dioxide (NO$_2$) limit values are still regularly exceeded in Europe, particularly in cities and along streets. This study explores how concentrations of nitrogen oxides (NO$_x$ = NO + NO$_2$) in European urban areas have changed over the last decades and how this relates to changes in emissions. To do so, the incremental

approach was used, comparing urban increments to total emissions and roadside increments to traffic emissions. In total, nine European cities were assessed. The study revealed that potentially confounding factors like the impact of urban pollution at rural monitoring sites through atmospheric transport are generally negligible for NO$_x$. The approach proves therefore particularly useful for this pollutant. The estimated urban increments all showed downward trends and for the majority of the cities the trends aligned well with the total emissions. However, it was found that factors like a very densely populated surrounding or

local emission sources in the rural area such as shipping traffic on inland waterways restrict the application of the approach for some cities. The roadside increments showed an overall very diverse picture in their absolute values and trends and also in their relation to traffic emissions. This variability and the discrepancies between roadside increments and emissions could be attributed to a combination of local influencing factors at the street level and different aspects introducing inaccuracies to the trends of the used emission inventories, including deficient emission factors. Applying the incremental approach was evaluated

as useful for long-term pan-European studies but at the same time it was found to be restricted to certain regions and cities due to data availability issues. The results also highlight that using emission inventories for the prediction of future health impacts and compliance with limit values needs to consider the distinct variability in the concentrations across but also within cities.

## 1   Introduction

The assessment of air quality has been of interest for many years because it is affected by human activity and simultaneously has an effect on us and our environment. Depending on the air pollutant, impacts on human health, ecosystems and the climate


have been observed (Guerreiro et al., 2014). In this study the focus is on $NO_x$, i.e. the sum of nitric oxide (NO) and nitrogen dioxide ($NO_2$). Even though limit values are defined for $NO_2$, the two pollutants are often assessed as $NO_x$ due to their close chemical coupling (Carslaw and Beevers, 2005). The share of either pollutant (NO and $NO_2$) in the total amount of $NO_x$

depends on many factors such as the initial emissions, ambient concentrations of ozone ($O_3$) and meteorological conditions (Carslaw and Beevers, 2005; von Schneidemesser et al., 2017). Due to the close coupling of NO and $NO_2$, emission inventories also typically report $NO_x$ emissions (Lee et al., 2015). Like most air pollutants, $NO_x$ can have adverse effects on human health and ecosystems. Exposure to $NO_2$, both short-term and long-term, has been associated with increases in all-cause mortality as well as with adverse effects on cardiovascular and respiratory systems (Faustini et al., 2014; Mills et al., 2015). Furthermore,

$NO_x$ plays an important role in the formation of secondary pollutants such as particulate matter (PM) and $O_3$ (Carslaw et al., 2011a; von Schneidemesser et al., 2017), which in turn are associated with adverse effects on health, ecosystems and climate (Guerreiro et al., 2014; Richter, 2009).

    In the European Union (EU), limit values for $NO_2$ in the atmosphere are in place since in 1985 (Council of the European Union, 1985). The most recent limit values are defined in a directive from 2008 and are set to an annual mean of 40 µg m$^{-3}$

and an hourly mean of 200 µg m$^{-3}$ that should not be exceeded more than 18 times a year (European Parliament and Council of the European Union, 2008). These limit values are still regularly exceeded. In 2017, exceedances of the annual $NO_2$ limit were recorded in 16 EU Member states (European Environment Agency, 2019). Urban areas are particularly affected and in 2017, 86 % of the exceedances of the annual limit value were observed at air quality monitoring stations along roads (Carslaw et al., 2011a; European Environment Agency, 2019).

Another aspect that makes $NO_x$ a pollutant of particular interest is the discrepancy that was found between emission standards and real-world emissions from some vehicle classes (Vestreng et al., 2009). Since the 1970s, road transport has been the main source of $NO_x$ emissions and in recent years it was found that these emissions did not decrease as expected (Vestreng et al., 2009). To reduce air pollution, so called Euro emissions standards were introduced in the 1990s which set limits for the exhaust emissions from new vehicles (Font and Fuller, 2016). These standards have been gradually tightened over the last

decades but it was noticed that the $NO_x$ emissions under real-world driving conditions did not reduce accordingly for some vehicles (Carslaw et al., 2011a). Especially diesel vehicles were found to have higher emissions (Carslaw et al., 2011a). It was estimated that if true on-road $NO_x$ emissions of diesel vehicles met the regulatory standards this would result in a reduction of the overall traffic emissions in a city by 30 to 75 % (von Schneidemesser et al., 2017).

    Comparing emission inventories (EIs) to measurements can help in understanding any inadequacies and uncertainties that

are related to the EIs (Beevers et al., 2012). It has been acknowledged that even though the quality of EIs improved over the last years there are still large uncertainties associated with them (Lee et al., 2015; Vestreng et al., 2009). One way of evaluating EIs through a comparison with concentration measurements is by analysing trends rather than absolute or modelled values. When analysing long-term trends aspects like chemistry and meteorology can be assumed negligible and it can be assessed whether the changes in the observed ambient concentrations reflect the changes that would be expected based on the emissions

(Beevers et al., 2009).





The comparison of the trends in measurements and emissions was done using the so called incremental approach. It was first used in a study by Lenschow et al. (2002) on PM in Berlin and is therefore also referred to as the Lenschow approach (Thunis, 2018). According to the approach, the difference between urban background and rural background concentration levels of a pollutant is the urban increment and similarly, the difference between measurements from urban traffic sites and the urban
background makes up the roadside increment. According to Lenschow et al. (2002), the urban increment can be attributed to the total emission sources of a city and likewise, it can be assumed that traffic emissions lead to the roadside increments (e.g. Beevers et al., 2009; von Schneidemesser et al., 2017). Following from this, it is then assumed that any changes in either of the emissions should lead to proportional changes in the related increment, again in line with the above-mentioned studies.

A major advantage of this approach is that any large scale processes, such as meteorological conditions or the dynamics
of the mixing height of the atmosphere, should not confound the analysis because they are influencing both urban and rural concentration levels (Font and Fuller, 2016). Especially variations in the meteorological conditions can either falsely emphasise or mask a trend in the concentrations and so minimising this effect allows a better comparison with emission changes (Beevers et al., 2009). Nevertheless, there are also aspects that have been criticised about this approach. For example, Thunis (2018) and Torras Ortiz and Friedrich (2013) noted that it is assumed that the rural measurements are not impacted by urban emissions
even though this can likely be the case. To what extent this affects the accuracy of urban increments was evaluated before using the approach in this study.

The incremental approach is well established for the analysis of PM, but has seen less application yet for the analysis of $NO_x$. Urban increments of PM and $NO_2$ were for example calculated by Elser et al. (2016), Petetin et al. (2014), Pey et al. (2010) and Thunis (2018), and roadside increments of PM, $NO_2$ and $NO_x$ were estimated for example by Beevers et al. (2009), Font
et al. (2019) and von Schneidemesser et al. (2017). Those studies all focus on individual cities, e.g. Berlin (von Schneidemesser et al., 2017), or small groups of cities, e.g. London and Paris (Font et al., 2019), and mainly assess ambient concentrations in major urban areas. Furthermore, most of the studies analyse set periods of time, such as one year (Elser et al., 2016), five years (Beevers et al., 2009), or ten years (Font et al., 2019). In contrast to that, the study at hand attempts to apply the approach systematically to the analysis of trends in $NO_x$ across all of Europe. On one hand, this means that also smaller urban areas
are included in the analysis. On the other hand, it also enhances the temporal scope to over 25 years for some cities instead of limiting it to a set time period.

The aim of this study is to examine how $NO_x$ concentrations and emissions have evolved over the last decades in European cities, and how their trends relate to each other. Changes in urban and roadside increments will be compared to changes in total and traffic emission inventories to assess whether the trends align as would be expected or whether any discrepancies can be
found. Furthermore, the study aspires to evaluate if the incremental approach is a useful tool for pan-European studies on $NO_x$, especially with regards to data availability and aspects that are sometimes criticised about this approach. This was done by an examination of the data that is available for European cities, assessing the incremental approach in detail and finally applying it to the cities where the data allowed it.



## 2 Data and Methods

Except as stated otherwise, all analyses were carried out using RStudio (RStudio Team, 2019) and R (R Core Team, 2020), along with the R packages listed in the supplement (Table S1).

### 2.1 Data

The measurements of ambient $NO_x$ concentrations were obtained from the AirBase database. Observations from rural background (RB), urban background (UB) and urban traffic (UT) stations were used to calculate urban and roadside increments.

From the respectively classified stations, daily $NO_x$ data was used to calculate annual averages for all years with sufficient data between 1990 and 2017. The data had to meet the following criteria, as suggested by the European Environment Agency (2009): the annual coverage of values should be at least 75 % in order to include data from the respective year, and each individual record needed to pass a visual quality screening. No minimum record length was set for urban stations.

The emission inventories (EIs) were sourced as gridded emissions from the European Monitoring and Evaluation Programme

(EMEP). The EMEP grid provides annual emission data from 1990 to 2017 in a $0.1° \times 0.1°$ long-lat resolution in the geographic coordinate system WGS84 (Wankmueller, 2019). It is based on the national and sectoral emissions that are reported by the participating states under the Convention on Long-range Transboundary Air Pollution (CLRTAP). Using QGIS 3.10 (QGIS Development Team, 2019) and shapefiles of the city boundaries (sources in the supplement, Table S2), city averages of total and traffic $NO_x$ emissions per year were calculated from the gridded EIs.

Data on wind directions and wind speed was obtained for seven cities from the respective national weather services. For most cities, hourly data was available, but for two cities only three measurements per day were provided. A table listing the measuring stations that recorded the wind data as well as the temporal resolutions and time periods of the data sets can be found in the supplement (Table S3).

### 2.2 Selection of Cities

The selection of cities was based solely on the availability of air quality observations. A search algorithm was developed to find all cities with at least one UB and two UT stations. The cities were sorted by the record lengths of their stations, and the ones with promising data availability were subject to a visual inspection of the continuity and consistency of the annual data of all respective UB and UT stations. The resulting preselection of cities was analysed for the availability of rural stations in their surroundings. For the RB, long records from single stations were not considered as important as for the other two station types, 115 but rather that throughout the whole time period any station provides data. This led to a final selection of cities that seemed suitable for the planned analysis.

### 2.3 Calculation of Increments

To analyse the air quality observations, increments were calculated based on annual averages derived from daily data. The urban increment was set as the difference between the UB concentration and the RB concentration (e.g. Lenschow et al.,





2002; Thunis, 2018; Torras Ortiz and Friedrich, 2013). Similarly, the roadside increment was defined as the UT concentration minus the UB concentration (e.g. Beevers et al., 2009; Font et al., 2019; von Schneidemesser et al., 2017). While the data of each UT station was used individually to calculate roadside increments, one representative urban as well as rural background concentration was established for each city. For the UB concentration this was done by averaging the annual data of all UB stations in a city. Similarly, the RB concentration was set as the average of the annual data from several RB stations. To do

so, the data of all RB stations which are within a 200 km radius around a city centre and which provided data for more than eight years was examined, and stations were selected based on their data ranges. Where the stations in the vicinity of the city showed relatively consistent concentration levels to each other, a maximum radius for including stations was defined as the distance where a deviation from the consistent level was observed. Furthermore, some stations very close to or still within a city were excluded if they showed distinctively higher values, along with some stations further away where the concentration

ranges indicated a potential local influence. Where the stations in the vicinity showed no consistent level, the city was excluded from the analysis of the urban increment.

Including stations with different record lengths in the calculation of a mean RB concentration can lead to inconsistencies in the number of stations providing data per year. In some years, only very few stations may have recorded data and if these stations measure comparatively high (or low) concentrations, such biases in the monitoring network can introduce misleading

trends in the average (Lang et al., 2019). To resolve this issue, the mean RB levels were calculated using moving window regression as suggested by Lang et al. (2019), with the therein recommended window width of three years.

### 2.3.1 The Impact of Atmospheric Transport

One of the underlying assumptions of the incremental approach is that the rural background concentrations are independent of the urban concentrations. If high air pollution in the urban area would lead to increased concentrations in the surrounding rural

area the calculated urban increment would underestimate the contribution of the city to the total concentration level (Pey et al., 2010). Some studies note that this is likely the case and that the urban increments should therefore be regarded as minimum (Pey et al., 2010; Thunis, 2018). A way in which urban pollution might impact rural concentrations is through atmospheric transport of pollutants from the city to rural areas (Torras Ortiz and Friedrich, 2013). To examine this effect, the daily RB data was split into subsets based on wind directions.

For each of the RB stations around a city, the direction in which the city is relative to the station's location was calculated, comparable to wind directions (0–360°). Based on (seven-)hourly measurements of wind speed and wind directions, daily averages of wind directions were calculated using vector calculations as denoted by Grange (2014). Afterwards, the daily data from each RB station was filtered by wind directions and split into two subsets: The first comprised of all days when the wind blew within a 30° range from the city to the station, and the second comprised of all other days. Hereinafter, the first subset will

be referred to as the "downwind days" and the second as "upwind days". For both subsets and each RB station, annual averages were calculated and subsequently the data from all stations around a city was averaged to two distinct mean RB levels. These downwind and upwind concentrations were compared with each other, as well as with the RB level based on the data from all days.





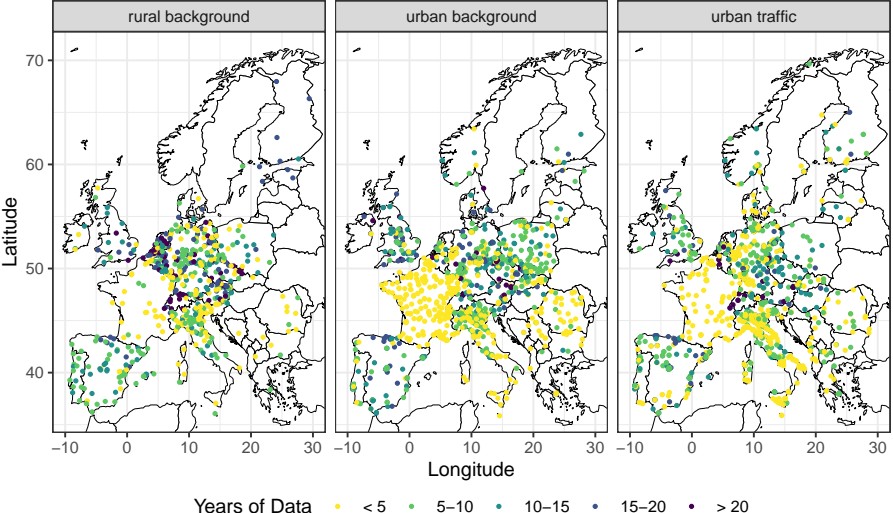

**Figure 1.** Air quality monitoring stations from which $NO_x$ data is available through the AirBase database, coloured depending on the number of years that the provided records span. The stations are classified based on their locations as rural background, urban background or urban traffic sites.

## 3   Results and Discussion

### 3.1   Availability of Observational Data

The availability of monitoring stations measuring $NO_x$ shows great variability across different countries and regions of Europe. In Fig. 1 the locations of RB, UB and UT stations which are included in the AirBase database are shown and the sites are coloured according to the number of years for which they provide $NO_x$ data through the database. This reveals regional differences in the available number of stations as well as in the length of recorded time series. A high density of stations from 160   all three types is found in Central Europe while in Scandinavia, the Baltic states and Finland the numbers are rather low. In general, there are fewer rural than urban stations and this discrepancy is especially clear for France and most Eastern European countries. France also stands out when looking at the record lengths that the stations provide since practically none of them has more than five years of data available. Most sites providing more than 15 years of data are located in Central Europe, Great Britain and Spain. While only a fifth of all stations are RB sites the share of sites with records of 15 years or more is much 165   higher for this type: 26 % as opposed to 13 % and 10 % for UB and UT stations, respectively. The regional differences in the data availability restrict the analysis of cities with long-term data from all three station types to certain areas in Europe. In total, nine cities were selected which are all located in Central Europe and Great Britain. These are: Amsterdam, Augsburg, Berlin, Geneva, Linz, London, Prague, Vienna and Zurich.

For some countries longer datasets seem to be available, but are not provided in the AirBase database. For example, Font 170   et al. (2019) conducted a study on air pollution in London and Paris where they used ten years of $NO_x$ data – also for Paris where




**Table 1.** Distance ranges within which rural background (RB) stations were considered to show consistent ranges, the outliers that were removed, and the final number of selected sites. "-" denotes that no lower limit was set or no outliers were removed.

| City | Lower limit [km] | Upper limit [km] | Removed outliers | No. of sites |
|---|---|---|---|---|
| Augsburg | - | 124 | one value > 30 $\mu g\ m^{-3}$ | 6 |
| Berlin | 25 | 154 | - | 11 |
| Geneva | - | 80 | - | 3 |
| London | - | 190 | one value > 50 $\mu g\ m^{-3}$ | 7 |
| Prague | - | 100 | station CZ0UVSE; one value > 35 $\mu g\ m^{-3}$ | 12 |
| Vienna | 16 | 75 | - | 14 |
| Zurich | - | 125 | - | 8 |

only less than five years of data per station was available in the database. For Austria the number of stations for which data is provided through AirBase dropped drastically from 2012 to 2013 even though according to the Austrian Environment Agency all stations are still monitoring air quality up to date. Similar issues and findings with regards to the data availability through the database were also described in other studies and for other pollutants. For ozone, like for $NO_x$, it was found that there is an

uneven spatial distribution of stations with certain record lengths and that there are generally less sites in Northern and Eastern European countries and more sites in Central Europe and Great Britain (Chang et al., 2017; European Environment Agency, 2009). It was also noted for ozone that longer datasets do exist outside AirBase and are available through other sources for some countries (European Environment Agency, 2009). Hence, these issues do not only apply to $NO_x$, and it would be desirable to include all existing time series of air pollutant measurements in Europe in one database to allow long-term analyses which are

not geographically biased or restricted to certain regions.

## 3.2 Urban and Roadside Increments

### 3.2.1 Urban Increments

The mean UB levels for the nine cities were calculated based on all available UB data. For the mean RB levels rural stations were selected based on their distance to the city and data ranges. Table 1 lists the lower and upper limits for selecting RB sites

as well as any outliers that were excluded before calculating averages. The RB data around Amsterdam and Linz did not show any consistent levels in the vicinity of the cities. These two cities were therefore considered unsuitable for the calculation of a mean RB level and with that also unsuitable for the analysis of an urban increment (UI).

For the seven cities where a consistent rural background level was found, UIs were calculated. In the following, all increments as well as emissions will be presented both in absolute values and as percentages relative to a baseline year which was defined

for each city based on the available data. The absolute UIs are shown in Fig. S1 in the supplement and ranged between





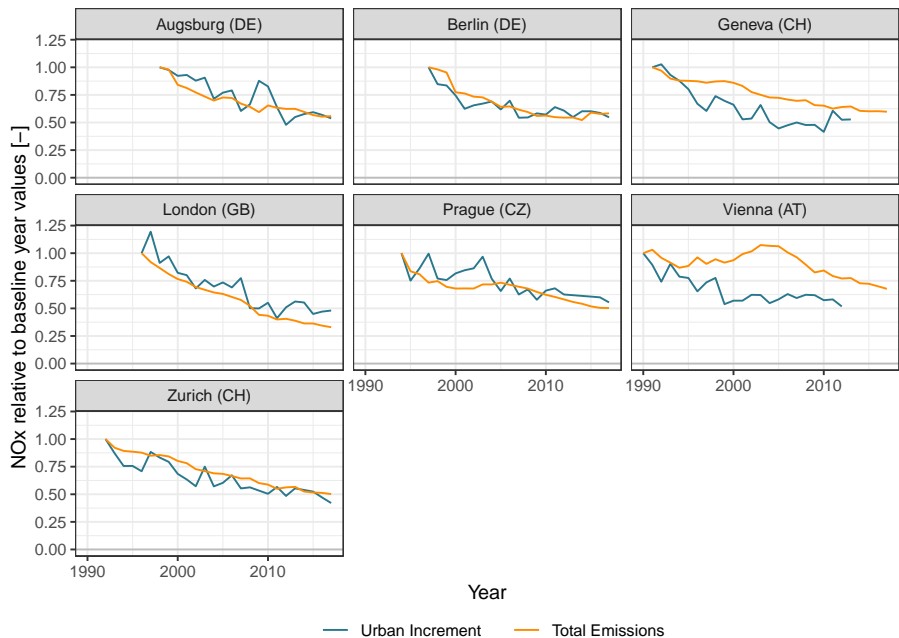

**Figure 2.** Urban increments (UIs) and total emissions of NO$_x$ for seven European cities, relative to their values from a baseline year which was defined individually for each city dependent on the data availability. The UIs are derived from rural and urban background monitoring stations; emissions are averaged for the urban areas based on the EMEP emissions grid.

23.34 ± 3.62 µg m$^{-3}$ in 2017 in Prague and 107.7 ± 48.8 µg m$^{-3}$ in 1997 in London. The percentage changes relative to the respective baseline years are shown in Fig. 2. The smallest reduction from the first to the last year with data was observed in Prague where the UI decreased by 44.6 % from 1994 to 2017. Declines between 45 and 49 % over twenty year periods were estimated for Berlin, Augsburg, Geneva and Vienna. In London the UI decreased by 52.0 % from 1996 to 2017 and in Zurich it decreased by 57.9 % between 1992 and 2017.

The absolute levels of the UIs exhibited a great variability. Torras Ortiz and Friedrich (2013) stated that it can be expected to find such variability when analysing UIs across different European countries due to the heterogeneity with regards to meteorology, topography and emission release conditions across Europe. Since most of the analysed cities are located in Central Europe the meteorological differences are expected to be negligible at the assessed time scales. The other two factors are assumed to be of great importance, especially in the cities with high UIs. The high levels in London can most likely be linked to high emissions, as London has by far the highest total emissions per square kilometre out of the analysed cities (see section 3.3). The second highest UI values were observed for Geneva. Here, high emissions are unlikely to be the key factor leading to the observed high concentrations. In Geneva, the topography plays a more important role. The city is located at around 400 m a.s.l. at the end of a valley which opens up to the north-east where Lake Geneva is situated. The remaining sides are framed with mountains which are between 1000 and 1700 m a.s.l. Cities in valley locations like this are known to show higher concentrations of





air pollutants because they are prone to the formation of stable inversions, especially in winter, leading to an accumulation of pollutants near the surface (Schultz et al., 2017).

The change over time of the UIs is characterised for all cities by a downward trend and reductions of around 50 % throughout the analysed periods. Some cities, e.g. Zurich, show a relatively consistent decrease over time while for other cities a stronger

decline followed by stabilising values was observed. The latter was the case in Vienna where the UI started to stabilise around 2000, and in London where the last ten years showed no strong trend. Studies on the trends of $NO_x$ measurements from different station types in the UK and London have found similar patterns: they observed a period that was characterised by decreasing concentrations followed by a period of more stable values (Carslaw et al., 2011a, b). In those studies, the change to more stable concentrations was observed between 2002 and 2004 while in the study at hand the downward trend extends until 2008. This

slight difference can be mainly attributed to the different lengths of the analysed records: the last years examined here were not yet included in the earlier studies.

The share of the pollution measured in a city that can be attributed to sources within the urban area was found to be similar for most analysed cities. It can be estimated using the ratio of the UI to the UB concentration and this ratio was calculated for each city and year. Overall, the ratios ranged between 55.9 and 84.5 % with a mean of 72.2 %. The highest average ratios

were found in London and Ausgburg: 77.5 %. In Berlin, Zurich and Prague the average ratios were between 70 and 75 % and in Geneva and Vienna they were around 65 %. The ratio is usually distinctively higher for $NO_x$ than it is for PM, where only around half of the pollution measured in a city also originates in the urban area (Lenschow et al., 2002). This is due to the more local characteristics and shorter lifetime of $NO_x$ (Liu et al., 2016). The differences in the transport characteristics of the pollutants can affect the accuracy of the estimated UIs. Studies on PM criticised the characterisation of urban areas simply by

their administrative borders instead of their actual urban sprawl or the extent of their commuting zones because it could lead to an underestimation of the total urban impact (Thunis, 2018; Torras Ortiz and Friedrich, 2013). This underestimation was not found for $NO_x$, suggesting that taking commuting zones into account is of greater importance for pollutants that are easily transported in the atmosphere. When analysing cities with a great urban sprawl it is nevertheless of particular importance to asses the data ranges of all surrounding RB stations prior to estimating a mean level to ensure that the stations actually represent

rural rather than suburban conditions. By doing so the chances for underestimating the UI are drastically reduced.

For Amsterdam and Linz no UI could be calculated because no consistent RB level was found. For Amsterdam this is likely to be caused by the high urbanisation of the Netherlands. The Netherlands is considered one the most urbanised countries in Europe (Nabielek et al., 2016) and is also among the European countries with the highest population densities (Barrientos, 2019). The urban areas of the country are characterised by many small to medium-sized towns with relatively short distances

between them (Nabielek et al., 2016). Some of the RB monitoring sites around Amsterdam are therefore likely influenced by urban areas, resulting in the great variation of concentration ranges that was observed.

The surrounding of Linz is not characterised by a particularly high urbanisation. Here, the varying topography in Austria or the positioning of some stations close to local emission sources such as industrial sites were hypothesized as potential causes for the varying data ranges. A detailed examination of the locations of the RB stations revealed that within the first 125 km of

Linz all sites with a median greater than 20 μg m$^{-3}$ are located east of Linz along the river Danube. While the other stations





are spread in all directions around the city and are far from the river, the sites with higher values are all within a maximum of 3 km from the river bank. The Danube is an important shipping route, and inland waterway vessels were observed to emit great amounts of $NO_x$ (Pillot et al., 2016). Most vessels which are in operation nowadays use diesel engines without any emissions control because they have been in service and registered before the first European guidelines on emission limits for inland

waterway vessels were introduced (Pillot et al., 2016). The high concentrations observed at some of the RB sites around Linz are therefore most likely caused by emissions from vessels on the river and should therefore not be considered representative of rural conditions.

### 3.2.2    The Impact of Atmospheric Transport

The impact that urban pollution has on rural background sites due to atmospheric transport was assessed for the seven cities

with consistent RB concentrations by calculating two mean RB levels: one based on "downwind days" and one based on all other days. For five of the cities, the mean RB level based on the "downwind days" is higher than the mean based on all days (see Fig. 3). This indicates that $NO_x$ is likely to be transported by wind from the cities to the surrounding rural areas. However, excluding the respective days leads to RB concentrations that differ only marginally from the RB concentrations where all days were included. When looking at annual $NO_x$ concentrations, the influence of atmospheric transport from a city to the rural

areas was therefore found negligible.

For Augsburg and Geneva the results are not as clear. In both cases the mean RB levels based on the "downwind days" are in some years higher and in others lower than the averages of all days. A potential explanation for the values found in Augsburg is that some RB stations might also be influenced by atmospheric transport from a different near-by city. Munich is only 60 km away from Augsburg, and some stations might measure higher values on days when the wind is not coming

from Augsburg because they are then downwind of Munich. Around Geneva the atmospheric transport is likely to be impacted by the surrounding topography as the city is situated at the end of a valley with high mountains rising on three sides. One of the assumptions of the conducted analysis is that the wind direction observed in a city can be taken as fairly representative for a bigger area around the city. While this simplified approach seems to lead to meaningful results for most cities, a very mountainous rural area could lead to a violation of this assumption. Furthermore, there are only three RB stations around

Geneva that were considered representative for the mean RB level and the data of those three stations was found problematic also for other analyses due to their relatively large data ranges.

### 3.2.3    Roadside Increments

Roadside increments (RIs) were calculated for all nine selected cities, separately for each UT station. The time frames that were spanned by the increments differed greatly and ranged from six years in Amsterdam and Berlin to 27 years in Zurich

and Prague. Similarly great variation was also found in the values themselves (Fig. S2 in the supplement). Both the highest and lowest RIs were estimated for two different stations in London. The overall highest value was 327.1 µg m⁻³ in 1991 at Cromwell Road in Central London and the overall lowest RI was -8.669 µg m⁻³ in 2003 at Kentish Way in the south-eastern London Borough of Bromley. Apart from London, RIs above 150 µg m⁻³ were only calculated for stations in Berlin. A few





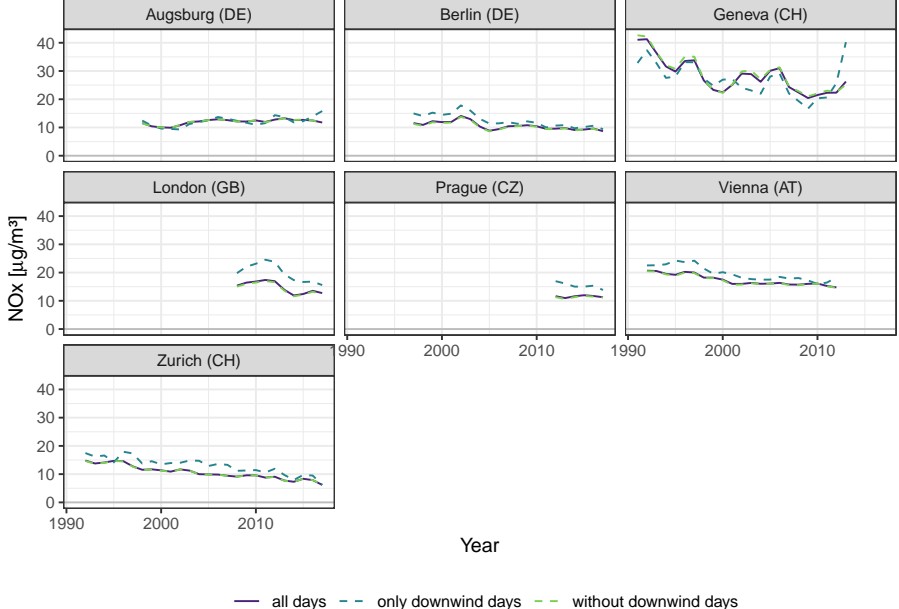

all days — — only downwind days — — without downwind days

**Figure 3.** The influence of atmospheric transport of $NO_x$ from cities to rural background (RB) stations. The depicted mean RB levels are based on three different subsets of daily data that were filtered by wind directions. "Downwind days" refers to all days when the wind blew within a 30° range from the city to a RB station.

years with values above 100 µg m$^{-3}$ were observed in Amsterdam, Augsburg, Prague and Zurich while the three remaining
cities had lower RIs.

In Geneva, London and Vienna negative RIs were calculated for some years at one station per city. A negative RI means that
the concentration at the roadside location was lower than the urban background level in that year. In London the reason for
this might be that the respective station is far from the city centre and that the calculated UB level is not representative for the
conditions in that area. In a city of the size of London more accurate RIs might be obtained when estimating two different UB
levels for the inner and outer city. In Vienna and Geneva, the observed negative RIs pose the question of whether the stations
are placed properly at representative locations. It was noted that the distinction between background and traffic stations in
Vienna is not as clear as for most other cities. Furthermore, different classifications of urban stations in Vienna according to
the AirBase metadata and information from the Austrian Environment Agency (2017) were found. The latter classification was
used.

Percentage changes relative to a common baseline year per city were calculated for the RIs (Fig. 4). The strongest percentage
change was observed at a station in Geneva where the RI decreased by 115.4 % from 1991 to 2014. The weakest reduction
was found at a site in Amsterdam: the RI reduced by as little as 9.4 %. However, this was at one of the stations with a record
of only six years, from 2012 to 2017. In most cities short periods or individual years with upward trends were observed, but
in Linz both UT sites were found to have increasing RIs throughout the recorded period. They increased by 49.2 and 64.7 %





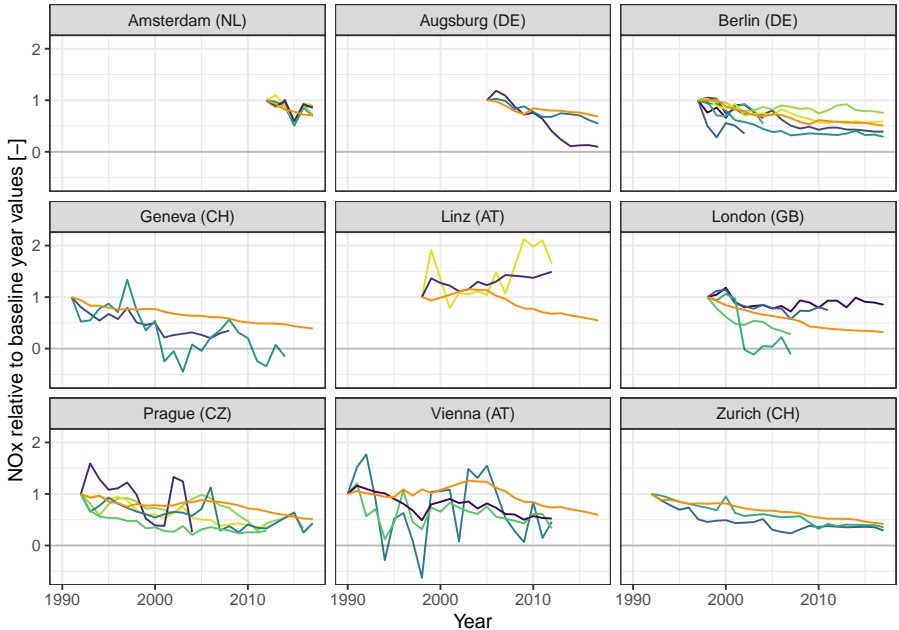

**Figure 4.** Roadside increments (RIs) and traffic emissions of $NO_x$ for nine European cities, relative to their values from a baseline year which was defined individually for each city dependent on the data availability. The orange lines represent traffic emissions and these are averaged for the urban areas based on the EMEP emissions grid. All other lines depict individual RIs which are specific to a certain urban traffic monitoring station.

from 1998 to 2012. Some UT sites in Amsterdam, Geneva, Linz and Prague showed very strong year-to-year variations in the percentage changes due to relatively low RIs in the respective baseline years. The differences in the absolute values at those sites were small with around 10 µg m⁻³, compared to the variations observed at other stations, e.g. in Berlin or London.

In most cities the direction of change agrees between the different RIs in the city. While the magnitude of change might differ, the sites from one city either show an upward or a downward trend. Only in some cities there are sites which were observed 295 to have rather stable values while other RIs from the same city continued to decrease. Great differences in the magnitudes of change were found, for example, in Augsburg and London. In Augsburg, one of the sites was at 54.6 % of the 2005 value in 2017 while the other station reduced down to 9.6 % in the same period. In London, at a station recording only ten years of data the RI decreased to -11.0 % of its 1998 value, while another station was still at 85.3 % even after a twenty year period.

The reasons for differences in the magnitudes of change of several RIs in one city can be diverse. In London, three of the 300 four sites providing data from 1998 to 2007 showed very similar declines over the ten-year period with regards to the absolute RIs: they all decreased by 75 to 85 µg m⁻³ (Fig. S2 in the supplement). Since they had measured very different levels in 1998, the percentage changes nevertheless showed the above described discrepancy which can be attributed to the very different characteristics of the locations of the UT sites. The station with high values and a smaller percentage reduction is situated in the inner city while the other is located in the fringe area of London. For the two stations in Augsburg the discrepancy in change





is present in the absolute and percentage values. The stronger decrease at one of the sites can be related to measures specific to that station: Around 2010, the respective street was subject to extensive reconstructions which aimed at minimising the individual traffic and at expanding and strengthening the public transport (Stadtwerke Augsburg Verkehrs-GmbH swa, 2013). Most RIs are assumed to show a mix of the influence of the general trend typical for the respective city and the effects of local measures and local street characteristics.

Several studies have analysed the data of the monitoring sites in London over the last years. A large heterogeneity in the values and trends of UT sites has been observed (Font and Fuller, 2016). For the period 2004/2005 to 2009, Font and Fuller (2016) found that on average the RIs increased by around 1 % $a^{-1}$, while Carslaw et al. (2011b) estimated downward trends in the $NO_x$ values from UT sites, and Beevers et al. (2012) found a decrease of around -2 % at streets in outer London and a small increase or stable values at streets in inner London. Even though the estimated trends vary, the different studies agree

that the trends in the analysed years were rather small. This shows that in years with only slight changes or even stable values the selection and number of sites which are analysed and the way in which the results are aggregated can lead to different conclusions with regards to the overall trends. It also highlights that for RIs a separate assessment of the data of individual stations can lead to more and different insights than calculating an average RI.

### 3.3    Emission Inventories

Total and traffic $NO_x$ emissions were averaged for each city based on the EMEP emission grid. Both showed overall declines from 1990 to 2017 in all cities even though in some cases periods of increases could be observed (Fig. S1 and S2 in the supplement). The total emissions were analysed for the same periods for which UIs were calculated, thus only seven of the nine cities and slightly different time frames for each city were assessed. The highest total emissions of 72.30 t $km^{-2}$ were observed in 1996 in London and the lowest in 2017 in Augsburg: 8.822 t $km^{-2}$. The percentages relative to the baseline year

values are depicted in Fig. 2. In Vienna, the emissions decreased by as little as 22.9 % from 1990 to 2012 and showed an increase between 1995 and 2003. In Geneva, Berlin and Augsburg, emissions reduced between 35 and 45 % over periods of 23, 21 and 20 years, respectively. Reductions of around 50% were observed in Prague from 1994 to 2017 and in Zurich from 1992 to 2017. The greatest decline was found in London: the emissions in 2017 were 67.0 % lower than in 1996.

   The traffic emissions were assessed for all nine cities, but for different time periods depending on the baseline year of the

RIs of each city. In Amsterdam the baseline year was set to 2012 and in the following six years the traffic emissions decreased by 29.1 %. A decrease of around 30 % was also found in Augsburg between 2005 and 2017. In this city, a slight increase of the emissions can be seen from 2009 to 2010. The same pattern is also present in the traffic emissions of Berlin, for the same years. Overall, the emissions decreased by around 50 % between 1997 and 2017 in Berlin. In Prague the emissions halved as well between 1992 and 2017, and in Geneva they decreased a bit more than half between 1991 and 2014. The other Swiss city,

i.e. Zurich, showed a reduction by 58.3 % between 1992 and 2017. The two Austrian cities showed a similar pattern as was already observed for the total emissions in Vienna: the emissions increased until 2003 and then decreased again after 2005. In Vienna, the emissions were a quarter higher in 2003 than in 1990. Overall, they decreased by 26.3 %. In Linz, the reduction





amounted to 32.3 % between 1998 and 2012. As was the case for the total emissions, the highest reduction in traffic emissions took place in London: a reduction by 68.3 % from 1998 to 2017.

Reductions in total emissions are linked, for example, to changes in the emissions from industrial sites in the cities or from the domestic heating sector (Senatsverwaltung für Stadtentwicklung und Umwelt, 2018). Reasons for the observed downward trends in traffic emissions are European emission standards and associated technology changes and end-of-pipe control measures (Crippa et al., 2016), updates of the vehicle fleets which in turn can be accelerated for example by scrappage schemes or low emissions zones (LEZs), the LEZs themselves, and local measures such as speed limits, road-specific vehicle restrictions,

reconfigurations of road networks or adaptive signalisation to reduce congestion (Bigazzi and Rouleau, 2017). It is difficult to attribute observed changes to specific measures and the efficacy of the measures in bringing the levels down would have to be evaluated for each city individually and is not the focus of this study.

    The trends in the emissions in Vienna and Linz did not show the continuous decline that was observed for the other cities. The peak between 2003 and 2005 was found to be also present in the national emission inventory and is linked to a large

extent to an increase of the amount of fuel that was exported in the tanks of vehicles (Umweltbundesamt Österreich, 2016). It was estimated that roughly a quarter of the traffic emissions were actually caused in surrounding countries, due to the export of goods and relatively low fuel prices (Umweltbundesamt Österreich, 2016). This export of fuel is expected to be negligible in the cities and adopting the upward trend of the national emissions to the trends in the cities might therefore lead to an overestimation of the actual change that was present in the traffic emissions of Vienna and Linz.

### 3.4    Comparison of Increments and Emissions

The increments of the $NO_x$ measurements were compared to the $NO_x$ emissions using the percentage changes of either one. RIs and traffic emissions were compared for all nine analysed cities while UIs and total emissions were only compared for the seven cities where a consistent RB concentration could be estimated.

#### 3.4.1    Total Emissions and Urban Increments

For the majority of the seven cities the percentage changes observed in the UIs aligned fairly well with the changes of the total emissions, as can be seen in Fig. 2. Apart from Geneva and Vienna, the general trends of the $NO_x$ emissions seem to be reflected in the UIs. The UIs showed stronger annual variations than the emissions did so the percentage change in the final years with data should be seen as momentary values and the changes throughout the entire period should be considered to draw more meaningful conclusions. In Augsburg and Berlin emissions and UIs aligned well for the whole twenty year period, and also

the percentage change values in 2017 differed only marginally between the two measures. In Prague the UI and the emissions were at around 55 % and 50 % of the baseline year values in 2017 and also generally showed similar declines throughout the analysed time frame. Despite the overall good agreement there were more years when the emissions had reduced by a greater share than the UI had than the other way around. In London the same was observed, but even more distinct: only in three years greater percentage reductions in the UI than in the emissions were found. However, the discrepancy was relatively constant

over time so that the change in the UI still seemed to follow the change in the emissions. In Zurich the opposite to what was





seen in Prague and London was found: during most years the percentage change of the UI indicated greater reductions than that of the emissions. Nevertheless, the trends seemed to align well also in this city. In 2017, the UI had decreased by 57.9 % and the total emissions by 49.8 %.

The two cities that showed relatively strong discrepancies between the temporal changes of the two measures were Geneva
and Vienna. In the final years the differences amounted to 12.7 and 25.3 percentage points, respectively, but were even greater for many previous years. In Geneva, a good agreement was observed only for the first two to three years. Then, the UI showed a much stronger decline than the emissions for a few years, followed by a period when the changes were similar but at very different levels. In the last years, the emissions continued to decline at roughly the same rate as before but the UI increased. In Vienna the discrepancy appeared even stronger. While the emissions started to increase at around 1995 the UI continued
to decline and when the emissions started to decrease in 2005 the UI stayed at a relatively constant level. In between, the percentage changes of the two measures differed by around 50 percentage points.

The more or less consistent discrepancies that were observed in London and Zurich can be related to the annual variability of the UIs. In both cities, the UIs showed strong changes between the first two years. In London for example, the UI increased by almost 20 $\mu$g m$^{-3}$ from 1996 to 1997 and decreased again in the following year. Just as the value in the last year is a momentary
value, so is the UI in the first year. If the dataset of London would have started one year later the value of the baseline year would have been distinctively higher and the percentage reductions relative to that value therefore lower, resulting in a better agreement with the change of the emissions. Similarly, using 1993 as the baseline year instead of 1992 in Zurich would lead to a lower baseline value and also a better agreement with the percentage change of the total emissions. These examples show that the exact level of the relative reductions depends on the values in the baseline year and thus on the definition of that year.
An agreement of how the percentage changes develop over time was therefore judged to be more important than an agreement in the exact percentage reductions for a specific year.

In Geneva and Vienna, a simple shift by defining a different baseline year could not resolve the discrepancies because these also extended to the change of the two measures over time. The peak in the emissions that was observed in Vienna is not reflected at all in the UI. In section 3.3 it was discussed that the main cause of this peak were emissions from exported fuel.
The discrepancy between the UI and the total emissions is therefore most likely based on differences between the emissions from used versus sold fuel in Austria. The discrepancy that was found for Geneva is likely related to inaccuracies in the UI rather than in the emissions. The UI of Geneva is – out of all seven UIs – the one that is based on the fewest monitoring sites, namely one UB station and three RB sites. Furthermore, the data measured at those stations differs in some ways to the data from comparable stations in and around other cities. However, the small number of stations makes it difficult to evaluate
whether the measured concentrations can be considered accurate and representative for the respective location types.

Overall, the results support the assumption that a reduction in the total emissions should lead to a proportional reduction in the UI and that the latter one can be attributed to the emission sources from the city, as was first stated by Lenschow et al. (2002). Apart from annual variations, long-term changes in total NO$_x$ emissions can give an idea about how the background concentrations in a city developed.





### 3.4.2 Traffic Emissions and Roadside Increments

The percentage changes of the RIs span a very wide range, also in their relation to the traffic emissions. For some UT sites the estimated RIs showed a stronger decline than the emissions while others reduced less. Similarly, there is a lot of variation between the cities, as can be seen in Fig. 4. In some cities the RIs and the emissions seem to align well, e.g. in Zurich, in others there is a great discrepancy, e.g. in Linz, and in yet others the different UT stations within a city show different signals and agree to varying levels with the emissions, e.g. in Berlin and London.

In Amsterdam, all five RIs showed a similar pattern of annual variations throughout the six years with data, with a distinct dip in 2015. The emissions declined relatively consistently during the same period. In 2017, two RIs had declined by 30 % relative to the 2012 values – just as the emissions had. The other three sites decreased by only 10 to 15 %. Overall, the reductions in the RIs and the traffic emissions seem to align for Amsterdam, but with just six years of data it is hard to judge. A very interesting feature in the traffic emissions of the city was observed from 2009 to 2010 when the values dropped by around 10 t km$^{-2}$. Unfortunately, not enough RIs could be calculated for that time to evaluate whether the drop in the emissions was reflected in the measured concentrations.

In Augsburg, the two RIs were found to have changed very differently after 2010: one decreased by around 55 % from 2005 to 2017 while the other reduced by over 90 %. The emissions showed a reduction of around 32 %. The trends in the percentage changes appeared to agree well between the emissions and the first RI even though the values in the final year differed. The other RI showed a much stronger decline than the emissions which – as was discussed before – could be related to reconstructions of the street network near the respective UT site. The RI is therefore considered to represent rather local changes in ambient concentrations and it is not surprising that these do not align with the changes of the average traffic emissions of the city.

Berlin had a wide range of percentage changes in the RIs: the values were at 29.0 to 74.5 % of their 1997 levels in 2017. Over the same period, the traffic emissions reduced to 51.0 % of their 1997 level. Three of the four long-term UT sites showed similar reductions until around 2005, subsequently diverging to different levels. Until 2005 the percentage reduction of the emissions was also in line with that of the three UT sites. The RI of the fourth station diverged earlier and ended at a percentage level well below that of the emissions. The stronger decline observed at that station is linked to a set of local measures: in 2000 a speed limit of 30 km h$^{-1}$ was introduced and an extension of an urban motorway parallel to the street was completed (Senatsverwaltung für Umwelt, Verkehr und Klimaschutz, 2019), and in 2005 HGVs weighing more than 3.5 t were banned from driving through the street with the exception of originating or terminating traffic (Senatsverwaltung für Stadtentwicklung und Umwelt, 2013).

The two RIs that could be calculated for Geneva showed similar trends and patterns in their absolute values but differed by around 50 μg m$^{-3}$ in the baseline year 1991. Due to their different starting levels their percentage changes also differed substantially in some years. Compared to the traffic emissions, both showed stronger percentage reductions. The site with higher RI values only provided data until 2008. By that time the RI had decreased by 65 % relative to 1991 while the emissions had reduced by only 41 %. In that year, the other RI showed almost the exact same percentage reduction as the emissions: 44 %. However, this was only momentarily because the RI peaked in that year. After 2008 the RI declined – even to negative





values – and in 2014 it was at -15 % of its 1991 value. The emissions were at 47 % of their baseline level in that year. Overall,
the RIs showed slightly steeper declines than the emissions especially in the first ten years and seemed to stabilise around 2005
while the emissions showed a continuous decline.

In Linz, a strong discrepancy between the RIs and the emissions was found, especially in the second half of the analysed
period. From 1998 to 2003 or 2005 the two RIs and the emissions showed similar increases. By 2003, the emissions had
increased to 116 % of their baseline value and the RIs to 105 and 115 %. In the following years they developed quite differently:
the emissions started declining and in 2012 they were at 68 % of the 1998 level. The RIs on the other hand kept increasing
to 165 and 149 % of their baseline levels. The two UT sites are located in very different areas of the city so that the upward
trends are unlikely to be caused by a local measure, but are rather linked to the general development of the city. Linz developed
from a "steel city" to an important service and cultural centre in Upper Austria and a vibrating, lively and growing city with
a high number of daily commuters from surrounding municipalities (Amt der Oberösterreichischen Landesregierung, 2018).
Furthermore, Linz is considered a trans-European transport nodal point (Amt der Oberösterreichischen Landesregierung, 2018).
From 2001 to 2011, the total number of journeys in the city increased by almost 18 % (Magistrat der Landeshauptstadt Linz,
2012). This strong increase is most likely what drove the observed increase in the RIs. The traffic emissions of Linz do not
show a comparable upward trend as the gridding of the national EI does not account for such local changes.

London is one of the cities where some RIs decreased more and others less than the emissions did. The two sites with
stronger percentage changes only provide data until 2007 and in that year the respective RIs are at -11 and 28 % of their 1998
levels. The emissions decreased to 58 % of their baseline level. The other two RIs are at 57 and 72 % of their 1998 values in
2007 and show relatively stable values or slight increases in the years afterwards. The only RI that could still be calculated for
2017 was at 85 % of the baseline value as opposed to the emissions which were at 32 %. In section 3.2.3, it was discussed that
one reason for the strong differences in the percentage changes of the four RIs are the different characteristics of the stations'
460   locations. The station where the RI showed negative values is situated in one of London's outskirts while the station with the
weakest decrease is in the inner city. London is the biggest of the analysed cities and therefore shows a lot of variation within
the city. Nevertheless, the emissions were averaged over the entire urban area to retain consistency with the other cities. To
compare RIs with emissions that are more representative of the area where a respective UT site is located it might be useful to
split all data into Inner and Greater London.

465   For Prague, many RIs could be calculated and most of them showed similar trends. In the first five years after 1992 the
decrease of the majority of the RIs agreed with the trend of the emissions. In the following years the RIs generally decreased
further – apart from annual fluctuations – while the emissions increased slightly until 2005 and then decreased again. In 2012,
four UT sites had provided data and the corresponding RIs had all reduced by around 60 to 70 % relative to 1992. The emissions
had only reduced by 34 % in the same period. In 2017, the only remaining RI had decreased by 56 % and the emissions by
470   48 % so that integrated over the entire period the percentage changes were almost identical. However, the way the RIs and the
emissions changed over time did not align very well and generally, most RIs showed stronger reductions for most of the time
than the traffic emissions did.





The traffic emissions in Vienna showed a peak around 2004, just as was observed for the traffic emissions of Linz and the total emissions of Vienna. The three RIs did not follow that trend. One RI also showed high percentage change values from 2003 to 2005 but this RI had rather low and partly even negative absolute values so that the percentage values fluctuated quite strongly. The other two RIs increased from 1998 to 2001 but then started decreasing while the emissions were still increasing. As was mentioned before, the emission peak would probably not be present if the EI excluded emissions from exported fuel and so the RIs were not expected to reflect that trend. In 2003, the emissions were at 126 % of their 1990 level and the RIs at 66, 86 and 149 %, respectively. In the last year with data, i.e. 2012, the emissions were at 74 % of the baseline value while the RIs had decreased to 33 to 52 % of their values from 1990.

Zurich shows a relatively good agreement between the percentage changes of the traffic emissions and one of the RIs even though the latter one stabilised in recent years while the emissions continuously declined. The other RI showed a stronger decrease until around 1998 and then started to level off or slightly decrease. From 2010 onwards the percentage changes observed in both RIs were very similar to each other and showed no distinct trend. By 2017, both had reduced by 65 to 70 % of their 1992 levels. During the same period the traffic emissions decreased by 58 %. The stronger percentage change of one of the RIs in the first years is to a large extent due to the lower absolute value in the baseline year. The stabilisation of the RIs after 2010 while the emissions kept declining could indicate that the amount of $NO_x$ that was actually emitted in the city did not decrease as suggested by the EI. Similar findings were described in other studies and related to the discrepancies between $NO_x$ emission factors and real-world emissions from diesel vehicles as will be discussed in more detail further down in this section.

Overall, the RIs did not align as well with the emissions as the UIs did. They are more locally influenced which is why even RIs from the same city can show very different values and trends. RIs can be impacted by the characteristics of the street, like its width or its orientation relative to the main wind direction, as well as by local measures, like restrictions for certain vehicles, the introduction of speed limits, or a reconfiguration of the road network. Differences between RIs in the inner city as opposed to the outskirts of a big city were observed in London. All these are factors which impact the ambient $NO_x$ concentrations very locally and which are not expected to affect the average traffic emissions of a city. Other measures, like the introduction of a LEZ, could potentially influence the concentrations at all UT sites within the zone and their effects should also be seen in the average emissions due to the larger areal scope of the measures. All policies that affect the entire city or at least large parts of it should – if they were effective – lead to reductions in the average traffic emissions that are reflected in the RIs. Rather local measures on the other hand are one of the reasons for the discrepancies that were observed in the trends of RIs and traffic emissions.

Another reason for a poor agreement between the changes of emissions and RIs is that the EIs did not necessarily represent the actual changes of traffic emissions in the city. Rather, the EIs represent the emission changes observed at a national scale, and only the absolute levels were downscaled according to local characteristics. Especially the case of Linz demonstrated how this can lead to distinct discrepancies between the changes of emissions and RIs. In a study on London, the London Atmospheric Emission Inventory (LAEI) and the National Atmospheric Emission Inventory (NAEI) of the United Kingdom were compared to measurements of ambient $NO_x$ concentrations (Lee et al., 2015). The study found "a much closer agreement"





of the measurements with the LAEI than with the NAEI and related that to the more accurate representation of traffic in London and thus more accurate traffic emissions. Another study on London also stated that using national EIs in analyses that focus on

local scales can increase the degree of uncertainty in the results (Beevers et al., 2012). For studies that focus on cities, more accurate and meaningful results could be obtained if local instead of national EIs were used. However, it was noted that it can be difficult to obtain such local EIs and that they are sometimes very scarce in the number of years for which data is available.

Many studies, especially in London and the UK, have found similar disparities between the trends of $NO_x$ and $NO_2$ emissions and ambient concentrations (e.g. Beevers et al., 2009, 2012; Carslaw et al., 2011b; Font and Fuller, 2016; Lee et al., 2015).

Even when different EIs that were based on various sets of emission factors or activity data were analysed none of them were found to agree sufficiently well with the changes in the concentration measurements (Beevers et al., 2012; Lee et al., 2015). Some compared better while others showed very poor agreement but overall it was noted that the estimated trends in the emissions were "too optimistic" compared with the observed measurement trends (Beevers et al., 2012). Using remote sensing data led to the conclusion that emission factors that were used in those EIs were generally underestimating the real

emissions, particularly from some vehicle classes (Carslaw et al., 2011b, a). The assessments indicated that the theoretical reduction of $NO_x$ emissions from diesel vehicles as it was required by stricter emission standards did not actually take place and that some newer passenger cars still emitted at the same level as the previous generation cars had (Carslaw et al., 2011b, a). This disparity between emission factors and real-world emissions was identified as the main cause for the stronger reductions in the emissions than in the observed concentrations (Carslaw et al., 2011a). In the study at hand, it is difficult to evaluate to

what part the observed discrepancies are related to false emission factors which do not reflect real-world emissions because of the previously discussed factors that might also play a role.

The observed disparities between the trends in the RIs and the traffic emissions are in most cases linked to a combination of all of the discussed factors like local differences within a city, a poor representation of changes specific to the city by the EI, and inaccurate emission factors for some vehicle classes. The importance of each factor and how strongly they affect the

individual RIs can vary from city to city and probably even from UT site to UT site. Nevertheless, the fact that changes in RIs are in most cases not directly proportional to changes in estimated traffic emissions has some important implications. If EIs are used for future projections of ambient concentrations of $NO_x$ in cities the large uncertainties associated with that have to be taken into account. Trying to forecast how the air quality will improve in cities by using the trends of national EIs could lead to massive biases as was shown particularly by the example of Linz and as was also stated by Beevers et al. (2012). Similarly,

assuming the emissions will reduce as indicated by the emission factors has proven to lead to a potential overestimation of the trends in the ambient concentrations (Beevers et al., 2009). The high uncertainty associated with emission factors should not be ignored when drawing conclusions about future concentrations of air pollutants based on EIs. Especially when trying to evaluate future health impacts or compliance with limit values at the street level the general uncertainties as well as the local differences within a city need to be considered to ensure an efficient management of air pollution that is caused by traffic.





## 4 Conclusions

Trends in ambient concentrations and emission inventories of $NO_x$ were compared using the incremental approach. Urban and roadside increments were calculated for nine European cities which were selected based on the availability of $NO_x$ measurements through the AirBase database. These cities are: Amsterdam, Augsburg, Berlin, Geneva, Linz, London, Prague, Vienna and Zurich. Furthermore, the potential impact that urban pollution might have through atmospheric transport on rural background (RB) concentrations was assessed to evaluate the accuracy of urban increments (UIs) of $NO_x$.

The analysis of the availability of $NO_x$ observations through the AirBase database revealed some distinct regional differences. Both the number of stations and the lengths of the provided data records were higher in Central Europe and Great Britain compared to any other part of Europe. Accordingly, the analysis of European cities was limited to this region. It was further restricted to very few cities due to short records, data gaps and temporally non-overlapping records of urban background (UB) and urban traffic (UT) sites. Particularly extensive data sets were available for Berlin and Prague where high numbers of UB and UT sites provided data records for twenty or more years, and records of the same lengths could also be compiled from the surrounding RB stations. While more data seems to be available outside the AirBase database the data provided there sets serious limitations to the analysis of long-term trends in increments in pan-European studies.

Assessing the impact of urban pollution on rural concentrations showed that atmospheric transport from cities to rural sites is negligible when annual averages of $NO_x$ are considered. On the days when the wind came from a city towards a RB station the measured concentrations were usually higher but excluding those values had almost no effect on the annual averages. This could be different if UIs were analysed on shorter timescales, e.g. daily, or for other pollutants, e.g. PM.

Analysing the UIs showed that the assumption of finding consistent UB and RB levels in and around an urban area is true for most cities. However, for two of the nine cities the RB stations in the surrounding area did not show consistent concentrations so that those cities were considered unsuitable for calculating UIs. The reasons for that were a very densely populated country with few actually rural areas around Amsterdam, and a clear impact of inland waterway vessels on the river Danube at some of the RB sites around Linz. For the remaining seven cities consistent concentration levels were found and UIs were calculated and compared to total emissions. The percentage changes of the two measures generally aligned well, as was expected. Only for two cities, i.e. Vienna and Geneva, the changes of emissions and UIs differed. The discrepancies could be attributed to the emission inventory in one case and to probable inaccuracies in the UI in the other case. Over the analysed time frames from 20 to 25 years the UIs and the emissions showed overall downward trends in all cities. The UIs decreased by 45 to 60 % and the total emissions by 23 to 67 %, but not continuously in some cities. The share of the $NO_x$ concentrations measured in a city that originates in the urban area was estimated using the ratio of the UI to the UB level. For the different cities, the ratio ranged between 55 and 85 % with a mean around 70 % which shows that a great part of the $NO_x$ air pollution in a city is caused by activities within the city itself. Measures for reducing $NO_x$ concentrations should therefore strongly focus on the emission sources within urban areas.

The analysis of roadside increments (RIs) revealed a great variability within and across cities both in absolute levels and trends. Most RIs showed general downward trends but in some cities increasing values were found. Furthermore, some RIs





decreased to negative values, raising questions about the adequacy of the locations and type classifications of the respective
UB and UT sites. The traffic emissions showed overall downward trends in all cities with increasing values for a few years
in some urban areas. The relation of changes in RIs to changes in traffic emissions was found to vary strongly, again within
as well as across cities. In some cases, the discrepancies between the trends in traffic emissions and RIs might be related
to inaccurate emission factors in the emissions inventories (EIs). However, there are also other important factors playing a
role. City averages of traffic emissions do not accurately represent the emissions at the street level, and furthermore, RIs are
impacted by very site-specific factors such as speed limits or changes in the near-by road network. Considering the trend in
a RI representative for how $NO_x$ concentrations generally changed at the street level of the respective city should therefore
be viewed critically. Nevertheless, RIs provide valuable insight into how certain methods for emission reductions impact the
ambient $NO_x$ concentrations at streets.

A more accurate comparison of trends in emissions with trends in increments could be achieved if EIs were available that
are more representative of the true urban emissions. Ideally, the EIs should not include emissions from exported fuel and the
temporal changes should be downscaled or adopted properly to local changes in emission sources. While these more local
EIs with a decent temporal resolution can be accessed for some cities, the current availability is limited for pan-European
studies. Similarly, also the provisioning of $NO_x$ observations limits an analysis to certain European regions. It would have
been interesting to include cities in other parts of Europe to assess how differences in meteorological conditions, e.g. between
north and south, or differences in how traffic volumes and vehicle ages changed, e.g. between east and west, would have
impacted the increments and their trends, but the available data did not allow for this. To be able to explore these aspects
in future studies access through the AirBase database to long-term records measured in cities across the whole of Europe
would be very valuable. While the incremental approach was evaluated as a useful tool, particularly for $NO_x$ where potentially
confounding factors were found to be negligible, its application to long-term pan-European studies is currently restricted due
to data availability.

*Author contributions.* TB conceived the study and guided the data analysis. NO prepared the raw AirBase data set. EM performed the
analysis and wrote the paper with contributions from each author.

*Competing interests.* The authors declare that they have no conflict of interest.

*Acknowledgements.* This work was hosted by IASS Potsdam, with financial support provided by the Federal Ministry of Education and
Research of Germany (BMBF) and the Ministry for Science, Research and Culture of the State of Brandenburg (MWFK). Ambient concentration measurements were obtained through the AirBase (EEA) repository. Gridded emissions data was obtained from the EMEP (CEIP).
Wind data was provided by CHMU, DWD, MeteoSchweiz, Met Office and ZAMG. E. Macdonald acknowledges the valuable discussions
and comments by E. von Schneidemesser.



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
