# Peer review of "A Comparison of Long-term Trends in Observations and Emission Inventories of NOx"

_Atmospheric Chemistry and Physics, 2020_

## Referee Comment (RC1) · Anonymous Referee #2 · 12 Dec 2020

McDonald et al describe their work on comparing observed long-term trends with NOx emission inventories in European cities in a well written manuscript. They use the incremental approach to assess how observations at roadsides compare to traffic emissions and how measurements in cities compare to the total city emissions over the past decade and beyond. While the urban increment aligned well with the city emissions, comparison of roadside increments and traffic emissions proved to be more difficult due to compounding factors like local influences not captured in inventories.

The overarching goal of the manuscript was to evaluate the feasibility of using available observations and inventories to come up with a consistent European wide method of assessing trends in NOx emissions and compliance with NO2 air quality standards. The authors have done an excellent job in gathering available information, treating

the data with great care and suitable techniques and highlighting the strength and shortcomings of the approach.

After addressing my minor comments below, I recommend publication in ACP.

General Comments:

The authors have done a very nice job in presenting their findings in detail and concluding on the pros and cons of their method. Since a major motivation for looking at NOx is its adverse health effects and the regular exceedances of air quality standards across Europe, this manuscript would benefit from a concluding paragraph on what has been learned from this study that can be useful to mitigate air quality exceedances.

Specific Comments:

p. 1, l1ff: please add one sentence to the abstract explaining what an increment is, otherwise your summary is hard to follow for readers not familiar with the method.

p.4, l115: Please state already here where the reader can find this final selection of cities, e.g. see section 3.1.

p.9, l208: Vienna shows a very distinct "bump" in its emissions in the mid-2000s. Can the authors comment on the cause of this? . . . - I see, just say see Section 3.3 :-)

p.11, l 285 Please add an explanation on how the "common baseline year" was chosen/calculated.

Technical Comments:

p.2, l33: "since in", delete "in"

p.12 Figure4: the legend is missing

---

## Referee Comment (RC2) · Anonymous Referee #1 · 15 Dec 2020

The manuscript reports interesting results on long-term trends in observed NOx in comparison with its emissions. The authors found downward trends for the majority of the cities, which aligned well with the total emissions of NOx and listed the limit of the approached adopted. The authors also report "an overall very diverse picture in their absolute values and trends and also in their relation to traffic emissions". The latter results reflect the weakness of the approach, which can also be said to the weakness of the study. To this reviewer, a revision is required to exclude the climate perturbation on long-term trend analysis such as Yao and Zhang (ACP, 20(2), 721-773). After this, the authors may gain robust conclusion, which allow the study scientifically publishable.

---

## Author Comment (AC1) · 22 Jan 2021

**General comment**

We would like to thank both referees for their constructive and useful comments, which helped us to improve the manuscript. We have carefully revised the manuscript considering all aspects that were mentioned in the comments. Here, we provide our responses. In each case, we have copied the referee's comments in bold, which are then followed by our responses in standard script. We consider the changes which were made in response to the referees' comments to have improved the manuscript and we hope the editor and referees find the revised version suitable for publication in ACP. We also append a marked-up version of the manuscript with the changes mentioned in our

responses to the referees. Added text is written in red while deleted text is crossed out.

**Response to Referee #1**

**The manuscript reports interesting results on long-term trends in observed NOx in comparison with its emissions. The authors found downward trends for the majority of the cities, which aligned well with the total emissions of NOx and listed the limit of the approached adopted. The authors also report "an overall very diverse picture in their absolute values and trends and also in their relation to traffic emissions". The latter results reflect the weakness of the approach, which can also be said to the weakness of the study. To this reviewer, a revision is required to exclude the climate perturbation on long-term trend analysis such as Yao and Zhang (ACP, 20(2), 721-773). After this, the authors may gain robust conclusion, which allow the study scientifically publishable.**

The discrepancies between the trends of roadside increments and traffic emissions that were found in our study are very unlikely to be caused by the effect of climate variability on ambient concentrations for several reasons. If the differences in the trends were caused because the roadside increments were impacted by climate variability this would mean that all roadside increments within one city should deviate from the emissions' trend in the same way. This is not the case though. In some cities (e.g. Berlin, see Fig. 1) we see opposing trends in the roadside increments, with some being above and others below the trend of the emissions. When looking at annual data, climatic trends can be assumed to be homogenous within a city and so even if there was a climate impact that could be excluded from the trends of the roadside increments it would affect all increments from one city in the same way and could therefore not diminish the discrepancy to the emissions' trend for all roadside increments.

Berlin_RIs_Climate.pdf

**Fig. 1.** Relative changes in roadside increments (RIs) and traffic emissions of NOx for Berlin, and climate data for the same period from station 00433 in Berlin.

Furthermore, the approach that we used in our study already ensures that the trends are not biased or misleading due to climate variations. As stated in the introduction, a major advantage of the incremental approach is that any large scale processes, such as meteorological conditions or the dynamics of the mixing height of the atmosphere, should not confound the analysis (Font and Fuller, 2016). Urban background concentrations and roadside concentrations within one city are affected by the same large scale processes and climatic variations, and calculating the difference between these concentrations – as is done for the roadside increment – therefore minimizes any climatic trend in the data.

Last but not least, the found discrepancies between the trends are not unexplained, but plausible reasons for the observed "very diverse picture" regarding roadside increments were given and discussed in the manuscript. As stated in the conclusion, the discrepancies are due to a combination of inaccurate representations of the emissions at the street level and the very local nature of roadside increments which are influenced by very site-specific factors such as speed limits or changes in the near-by road network.

Due to the above-mentioned reasons we do not see the need to revise the analysis. Instead, we added a paragraph that refers to the potential impact that climate perturbation can have on long-term trend analysis as discussed for example by Yao and Zhang (2020) and argue why this is not of relevance when using the incremental approach (p. 19, l. 529ff in the marked-up manuscript).

**Response to Referee #2**

**McDonald et al describe their work on comparing observed long-term trends with NOx emission inventories in European cities in a well written manuscript. They use the incremental approach to assess how observations at roadsides**

**compare to traffic emissions and how measurements in cities compare to the total city emissions over the past decade and beyond. While the urban increment aligned well with the city emissions, comparison of roadside increments and traffic emissions proved to be more difficult due to compounding factors like local influences not captured in inventories.**

**The overarching goal of the manuscript was to evaluate the feasibility of using available observations and inventories to come up with a consistent European wide method of assessing trends in NOx emissions and compliance with NO2 air quality standards. The authors have done an excellent job in gathering available information, treating the data with great care and suitable techniques and highlighting the strength and shortcomings of the approach. After addressing my minor comments below, I recommend publication in ACP.**

**General Comments: The authors have done a very nice job in presenting their findings in detail and concluding on the pros and cons of their method. Since a major motivation for looking at NOx is its adverse health effects and the regular exceedances of air quality standards across Europe, this manuscript would benefit from a concluding paragraph on what has been learned from this study that can be useful to mitigate air quality exceedances.**

The study showed that roughly three quarters of the NOx air pollution measured in cities originate from sources within the urban areas themselves. This makes clear that a strong focus of the mitigation of limit value exceedances has to be the reduction of NOx emissions from urban emission sources. Furthermore, we found distinct differences between emission inventories and roadside increments which should be accounted for when using emission inventories to evaluate future health impacts or compliance with limit values at the street level. Only by considering these strong local differences within cities an efficient management of air pollution in urban areas can be ensured. A paragraph stating these aspects was added to the conclusion (p. 21,

l. 591ff).

**Specific Comments:**

**p. 1, l1ff: please add one sentence to the abstract explaining what an increment is, otherwise your summary is hard to follow for readers not familiar with the method.**

We added a short explanation to the abstract.

**p.4, l115: Please state already here where the reader can find this final selection of cities, e.g. see section 3.1.**

We added a cross reference as suggested.

**p.9, l208: Vienna shows a very distinct "bump" in its emissions in the mid-2000s. Can the authors comment on the cause of this?...- I see, just say see Section 3.3 :-)**

We added a cross reference as suggested, but a bit further up in the manuscript (p. 7, l190).

**p.11, l 285 Please add an explanation on how the "common baseline year" was chosen/calculated.**

The common baseline year was defined individually for each city dependent on the

data availability. For the urban increments the baseline year was set to the first year with urban and rural concentration measurements available, and for the roadside increments it was set to the year when the majority of traffic stations started to provide data. This was briefly stated further up in the manuscript, but was added again on the suggested page for clarity.

**Technical Comments:**

**p.2, l33: "since in", delete "in"**

Changed.

**p.12 Figure4: the legend is missing**

The legend was added and the figure caption adjusted accordingly.

**Supplement:**

[revised manuscript text omitted]